# Enhancing Nitrogen and Phosphorus Removal by Applying Effective Microorganisms to Constructed Wetlands

**Xiaotian Li** [1], **Qizhong Guo** [2], **Yintang Wang** [1], **Junzeng Xu** [3,4,]*, **Qi Wei** [3,4], **Lina Chen** [4] and **Linxian Liao** [4]

[1]   State Key Laboratory of Hydrology-Water Resources and Hydraulic Engineering, Nanjing Hydraulic Research Institute, Nanjing 210029, China; xiaotianleehhu@163.com (X.L.); ytwang@nhri.cn (Y.W.)

[2]   Department of Civil and Environmental Engineering, Rutgers University-New Brunswick, Piscataway, NJ 08854, USA; qguo@rutgers.edu

[3]   State Key Laboratory of Hydrology-Water Resources and Hydraulic Engineering, Hohai University, Nanjing 210098, China; weiqi8855116@163.com

[4]   College of Agricultural Science and Engineering, Hohai University, Nanjing 210098, China; chenlina2001@163.com (L.C.); liaolinxian@hhu.edu.cn (L.L.)

*   Correspondence: xjz481@hhu.edu.cn

**Abstract:** Rainfall occurs frequently in South China and results in recurring of drainage/discharge of nitrogen and phosphorus-rich water from paddy fields, which may cause serious non-point source pollution of receiving waters such as rivers. Moreover, time intervals between individual rainfall events are short (often only several days). Thus, not only is the treatment of discharge water needed, but a more rapid form of treatment is desired as well. On the basis of the literature, constructed wetlands could remove nitrogen and phosphorous from paddy field drainage/outflow, and effective microorganisms (EM) could also be added to enhance the removals. A field experiment was conducted to demonstrate the wetland effectiveness and EM enhancement. The experiment was conducted from June to October in 2016. By applying EM to constructed wetlands, after 8 days, concentrations of total nitrogen (TN), ammonium nitrogen ($NH_4^+$-N), nitrate nitrogen ($NO_3^-$-N), and total phosphorus (TP) were reduced by 88%, 91%, 89%, and 50%, respectively. Within the first 4 days, TN and TP concentrations were reduced by 78% and 40%, respectively, with EM application, in comparison to 50% and 20%, respectively, without EM application, representing additional respective reductions of 28% and 20% by applying EM. The results from the field experiment indicated a significant improvement of phosphorus and nitrogen removals by applying effective microorganisms.

**Keywords:** effective microorganisms; constructed wetlands; nitrogen and phosphorus removal; paddy field drainage; stormwater treatment

## 1. Introduction

China is one of the world's water shortage countries. Rice paddies consume a large proportion of water resources and drain a large amount of water with high nutrient contents, resulting in the problem of non-point pollution, which can reduce the amount of available water resources. In general, nitrogen and phosphorus, through runoff from paddy fields, especially during the period after fertilization, are the main sources of non-point pollution in South China [1,2].

Constructed wetland is a common method for treating wastewater, and it is widely applied in different fields, such as agricultural and industrial wastewater treatment [3–8]. Constructed wetland technology, using the wetland ecosystem to absorb nitrogen and phosphorus from drainage,

is an effective way to cope with non-point source pollution [2,9]. It also has a good effect in terms of nitrogen interception and the retention of ammonia nitrogen, nitrate nitrogen, and phosphorus [3,10,11]. Aquatic vegetables, planted in wetlands, can improve the water treatment effect. They can reduce the flow rate of the receiving water and increase the residence time of the water [12]. Many kinds of aquatic plants can reduce the total contents of nitrogen and phosphorus in wastewater [13], because roots of aquatic vegetables can absorb nutrients from eutrophic water by the bacterial community in the root zone, which can degrade many kinds of pollutants [14,15].

Effective microorganisms (EM) technology was developed in the 1970s at the University of the Ryukyus, Okinawa, Japan. EM comes with a liquid form and consists of naturally occurring beneficial microorganisms, such as lactic acid bacteria, yeast actinomyces, and photosynthetic bacteria [16]. Namsivayam et al. [17] conducted a study that applied EM to treat sewages or effluents, pointing out that EM has the potential to improve the effectiveness of treatment of domestic wastes. Moreover, many studies have shown that EM technology can improve water and soil quality and enhance the growth, yield, and quality of crops [17–20].

A past study that focused on purification performance of nitrogen and phosphorus in slightly polluted landscape water by a combination of EM and submerged plants showed that the combination of EM and *Hydrilla verticillata* had a good purification effect on total nitrogen (TN) and total phosphorus (TP) in lightly polluted water, and the removal rates reached 70% and 97.3%, respectively [21]. The removal rates of TN and TP increased by 23.4% and 2.1%, and 23.4% and 41.3%, respectively, compared with the treatments by aquatic plants or EM alone [21]. Another past study, using aquatic crop wetlands to remove nitrogen (N) in water drained from paddy rice fields, showed that concentrations of ammonium nitrogen ($NH_4^+$-N), nitrate nitrogen ($NO_3^-$-N), and total nitrogen (TN) during the first two drainage events were reduced within 8–14 days of storing by 82.3–92.8%, 84.5–94.3%, and 74.9–92.4%, respectively [22]. Yet another study, treating the drainage water from paddy fields by aquatic vegetable wetland–ecological ditch system (AWDS), showed that TN and TP concentrations decreased significantly by 76.7% and 34.1%, respectively, after 12 days of abatement [23]. However, there are still no studies that have used both EM and surface wetlands jointly to treat polluted water.

The present study focuses on the concentration reduction rates of nitrogen and phosphorus pollution of wetlands with EM, as the concentration is a common indicator of water effluent/discharge quality. The study also included mass removal reduction rates of nitrogen and phosphorus pollution in the drainage water in wetlands. Moreover, the uptake functions of aquatic vegetables on nitrogen and phosphorus pollution in paddy field drainage were analyzed.

## 2. Materials and Methods

### 2.1. Wetland Layout

The study was conducted in 2016 at Kunshan Irrigation and Drainage Experimental Station, Jiangsu Province, China. The soil within the wetlands has a pH of 7.4 and is mainly composed of clay.

Groups of wetlands and their label names are shown in Table 1. Figure 1 shows the layout of wetlands. There were six surface flow-constructed wetland cells with lengths of 5 m and widths of 4 m, each separated by polyvinyl chlorid (PVC) plates. Water depth inside the wetlands decreased naturally from 50 cm to 10 cm during each experimental period (8 days). The type of soil at the top layer is hydragric anthrosol. Saturation soil water content for the top 0–20, 0–30, and 0–40 cm soil layers are 54.4%, 49.7%, and 47.8%, respectively [23]. Wetland cells were connected to each other by inlet pipes during the water filling process. The inlet and outlet pipes were fixed onto the surface of the wetlands and were sealed with rubber stoppers when not opened.

**Table 1.** Label name of each wetland group.

| Wetland Groups | Label Name |
| --- | --- |
| Water bamboo and arrowhead | W1 |
| Water bamboo and water spinach | W2 |
| Water bamboo and cress | W3 |
| EM 1 (no aquatic plants) | EM1 |
| EM 2 (no aquatic plants) | EM2 |
| EM 3 (no aquatic plants) | EM3 |
| EM water bamboo and arrowhead | EMW1 |
| EM water bamboo and water spinach | EMW2 |
| EM water bamboo and cress | EMW3 |

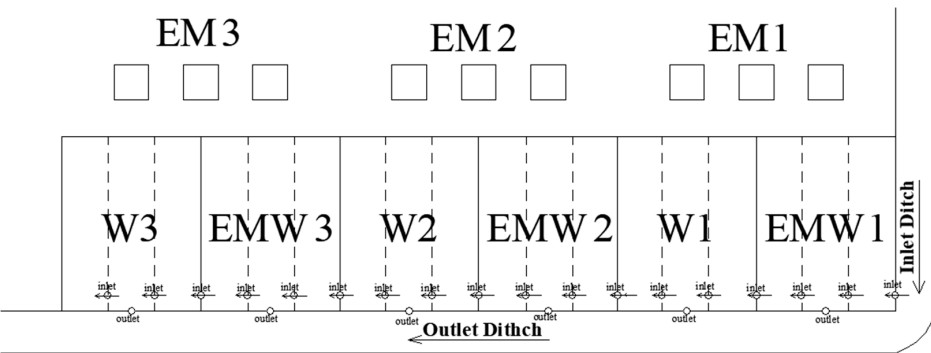

**Figure 1.** The layout of wetland cells.

Each wetland was divided into three pieces of small wetlands, planted with vegetable combinations. The non-EM wetlands (wetlands without EM been applied) and EM wetlands were planted with three separate vegetable groups, "water bamboo and cress", "water bamboo and water spinach", and "water bamboo and arrowhead". Wetlands were surrounded by PVC plates, which were buried 1 m from topsoil to prevent the exchange of water between different wetlands. Young plants of water bamboos, cress, and arrowheads and seeds of water spinaches were all planted on 5 May. In EM wetland (EMW) 1 and Wetland (W) 1, each small wetland was planted with one line of bamboo and one line of arrowhead. In EMW2 and W2, each wetland was planted with one line of water bamboo and one line of water spinach. In EMW3 and W3, each small wetland was planted with one line of water bamboo and one line of cress. In the harvest season, the total yield of vegetables was measured.

To simulate the effect of wetlands with EM but without planted vegetables, we divided nine boxes (1 m × 1 m × 1 m) into three groups, corresponding to small wetlands of "water bamboo and cress", "water bamboo and water spinach", and "water bamboo and arrowhead" groups. To ensure that the soil sample in the boxes were the same as that of the corresponding wetland, we took soil of each wetlands into the corresponding box until the depth reached 60 cm. Then, water was pumped from the corresponding wetland into the boxes until the water level in the box was the same as that of the corresponding wetland water layer. These three boxes were named "EM boxes" in this paper.

When paddy fields had surface runoff during storms, the water in the paddy field was directed into a ditch and then flowed into wetland cell EMW1 by gravity through the inlet pipe, and then flowed into rest of the experimental cells. Meanwhile, all the outlet pipes between wetland cells and the outlet ditch were closed at that time. All the inlet pipes were closed when the water level of each wetland cell reached about 30 mm. After 8 days of treatment, each outlet pipe, between the wetland cells and the outlet ditch, were opened to discharge the treated water.

Fertilizing dates of paddy fields were 5 July, 5 August, and 5 September. The experiment was carried out after the fertilizer was dissolved, typically 3 days after application. The experiment was designed to treat the water drained out from the paddy fields between rainfall events of short interval, and thus the treatment time duration was set to 8 days. Three durations for the three experiments

were selected for the field test: (1) 8 July to 16 July, (2) 8 August to 16 August, and (3) 8 September to 16 September. The paddy fields were fertilized three days prior to the start of each of the three field experimental periods.

### 2.2. Application of EM

EM was produced by a complete mix of clean water (80%), nutrient solution (10%), and primary EM (10%). To ferment and activate the EM, we produced the mixture in a closed container and kept it for 2 days at a temperature of around 30 °C. EM was applied in wetlands with a 0.001 volume ratio of EM to wetland water during three storm events.

### 2.3. Measurement and Analysis

Drainage water from the paddy fields stayed in wetlands for 8 days. Meanwhile, the water levels of wetland were measured at the beginning and the end of each experimental period. Then, we calculated the amount of water in the wetlands. Water samplings were obtained from each small wetland every 2 days, and then were filtered. For each sampling, water in three sampling points from a wetland was collected randomly and mixed evenly. The concentration of nitrogen was analyzed and averaged among three replicates to represent the value on that day. On the days when the water was drained into or out the wetlands, we also used the nitrogen concentration to calculate mass removal efficiencies.

Concentrations of TN, TP, $NH_4^+$-N, and $NO_3^-$-N in the water samples were determined by using alkaline potassium persulfate digestion spectrophotometric method, ammonium molybdate spectrophotometric method, Nessler's reagent colorimetric method, and the spectrophotometric method with phenol disulfonic, respectively [24]. Concentrations of nitrogen and phosphorus and changing volumes of wetlands were used to calculate nitrogen and phosphorous loads in wetlands. TN and TP concentrations in aquatic plants were measured using the Kjeldahl and acid melt-molybdenum stibium spectrophotometric methods [25].

## 3. Results and Discussion

### 3.1. Concentrations and Reduction Rates of Nitrogen and Phosphorus in Wetlands

The concentrations of TN in drainage water, for 8 days treatment, are shown in Figure 2, and the concentration reduction rates are shown in Table 2.

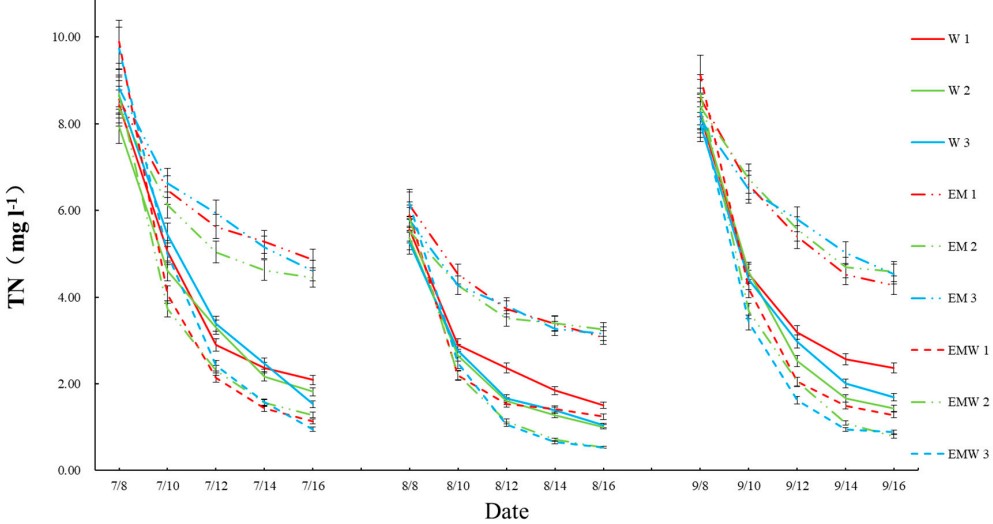

**Figure 2.** Total nitrogen (TN) concentration changes with time in wetlands.

**Table 2.** TN concentration reduction rates in wetlands over 8 days.

| Test No. | Time | Wetland 1 | Wetland with EM 1 | EM-Only 1 | Wetland 2 | Wetland with EM 2 | EM-Only 2 | Wetland 3 | Wetland with EM 3 | EM-Only 3 |
|----------|------|-----------|-------------------|-----------|-----------|-------------------|-----------|-----------|-------------------|-----------|
| 1 | 2 days | 40.16% | 58.98% | 24.43% | 42.08% | 57.08% | 26.71% | 37.14% | 48.69% | 24.86% |
| | 4 days | 65.64% | 78.36% | 34.26% | 58.38% | 73.46% | 39.74% | 60.89% | 75.19% | 32.68% |
| | 8 days | 75.26% | 88.48% | 43.14% | 77.10% | 85.26% | 46.76% | 82.32% | 90.22% | 47.77% |
| 2 | 2 days | 47.98% | 62.84% | 25.99% | 50.49% | 62.22% | 22.47% | 47.25% | 59.80% | 26.05% |
| | 4 days | 57.54% | 73.96% | 39.04% | 70.11% | 80.50% | 36.44% | 68.34% | 82.71% | 34.30% |
| | 8 days | 72.99% | 78.93% | 49.90% | 81.24% | 90.77% | 41.02% | 80.22% | 91.29% | 45.42% |
| 3 | 2 days | 44.61% | 54.08% | 23.49% | 44.87% | 57.71% | 19.84% | 44.92% | 58.73% | 19.89% |
| | 4 days | 61.12% | 77.60% | 37.32% | 69.61% | 76.38% | 33.65% | 62.79% | 80.40% | 28.41% |
| | 8 days | 71.13% | 86.00% | 50.27% | 82.75% | 90.93% | 45.36% | 78.78% | 89.25% | 44.01% |

In the whole three storage processes, concentrations of TN and TP in drainage water of wetland and EM boxes both decreased as time passed (Figure 2). The effect of EM on total nitrogen reduction was obvious in wetlands.

$NH_4^+$-N, $NO_3^-$-N, and TP concentrations of drainage water in wetlands for 8 days of treatment reduced similarly to TN concentration. $NH_4^+$-N, $NO_3^-$-N, and TP concentrations are shown in Figures 3–5. At end of 8 days of treatment, concentrations of TN, $NH_4^+$-N, and $NO_3^-$-N changed slowly.

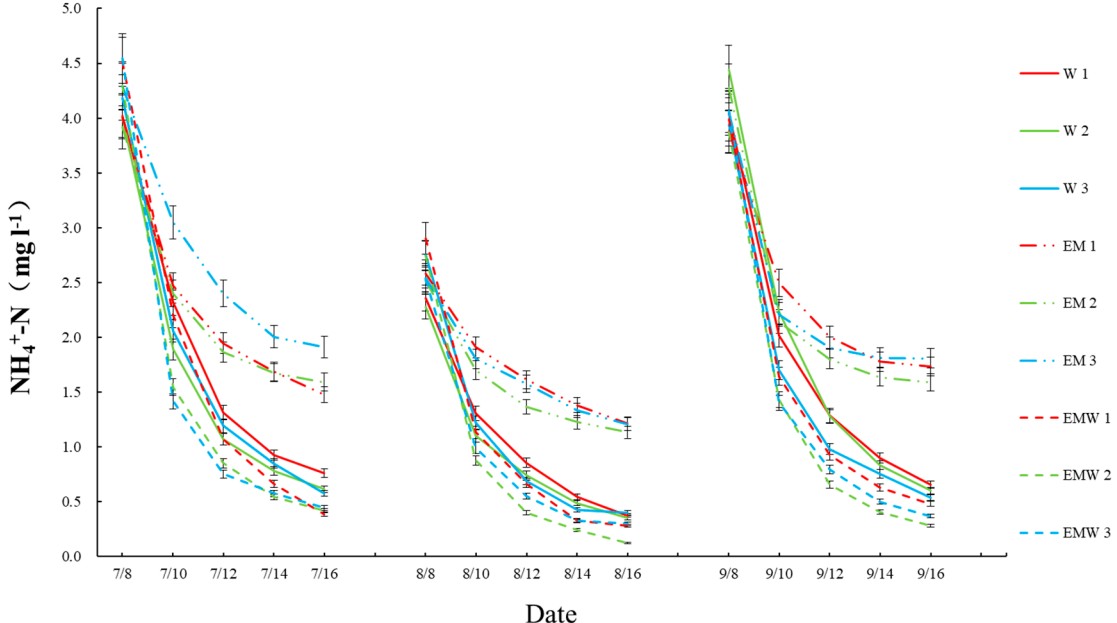

**Figure 3.** $NH_4^+$-N concentration changes with time in wetlands.

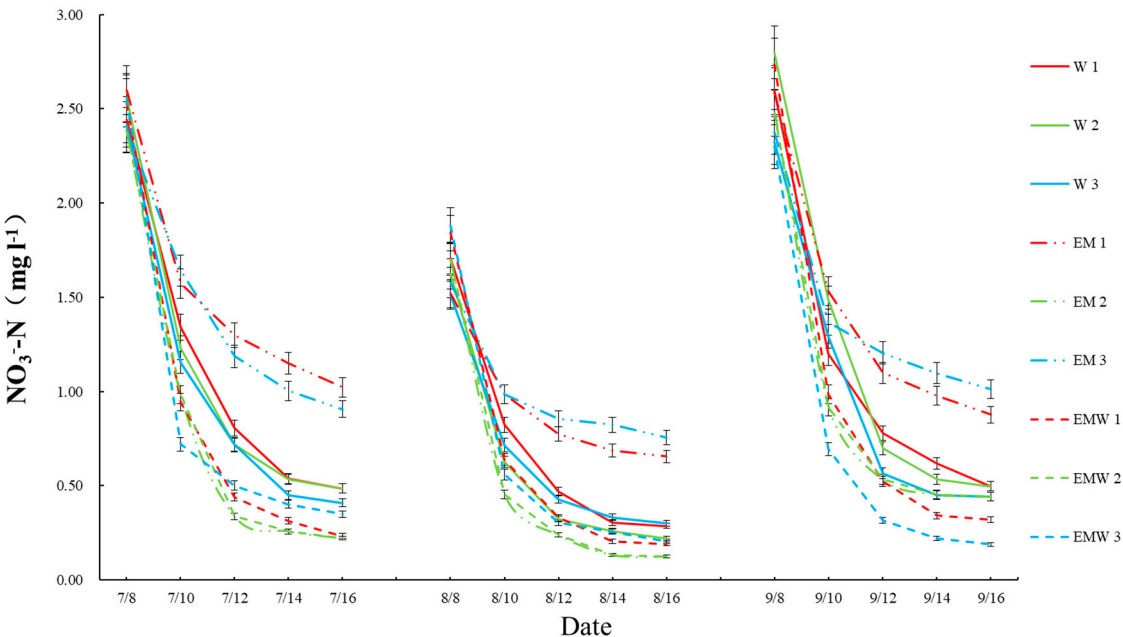

**Figure 4.** $NO_3^-$-N concentrations changes with time in wetlands.

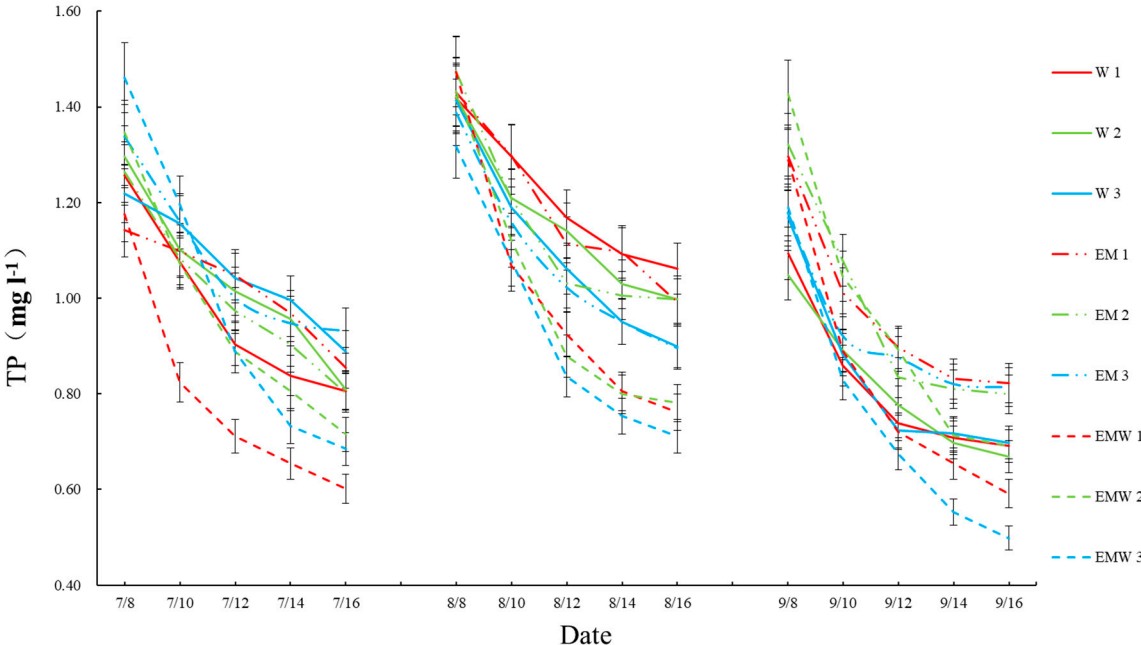

**Figure 5.** Total phosphorus (TP) concentration changes with time in wetlands.

The concentration reduction rates are shown in Table 2. The reduction rate is calculated from the difference between concentration at a later day (e.g., the 8th day) and the initial concentration (at the 0 day) divided by the initial concentration. The concentration reduction average rate of TN, in three wetlands with EM and different vegetable combinations, was 88%, higher than that in non-EM wetlands (78%) and that in EM boxes (46%). In particular, in the first 4 days, concentration of TN in EM wetlands with different vegetable combinations decreased sharply. Finally, the TN concentration of the drainage water in EM wetland were lower than that in the corresponding wetlands.

The reduction rates for $NH_4^+$-N and $NO_3^-$-N are shown in Tables 3–5.

$NH_4^+$-N and $NO_3^-$-N concentration reduction rates in non-EM wetlands could reach more than 80%, and TP reduction rates were above 30%. In EM-boxes, $NH_4^+$-N and $NO_3^-$-N reduction rates were around 48% and 52%, respectively; meanwhile, the TP reduction rate was around 25%. The treatment effect of EM wetlands resulted in higher reduction rates, with $NH_4^+$-N, $NO_3^-$-N, and TP reduction rates being 91%, 89%, and 50%, respectively, which were significantly higher than other groups of wetlands, especial in the first 4 days.

**Table 3.** $NH_4^+$-N concentration reduction rates in wetlands over 8 days.

| Test No. | Time | Wetland 1 | Wetland with EM 1 | EM-Only 1 | Wetland 2 | Wetland with EM 2 | EM-Only 2 | Wetland 3 | Wetland with EM 3 | EM-Only 3 |
|---|---|---|---|---|---|---|---|---|---|---|
| 1 | 2 days | 42.39% | 51.27% | 38.50% | 52.90% | 64.05% | 38.74% | 50.89% | 68.90% | 28.91% |
| | 4 days | 67.38% | 76.36% | 51.50% | 73.36% | 80.21% | 52.37% | 71.58% | 83.49% | 44.07% |
| | 8 days | 81.16% | 91.46% | 63.19% | 84.71% | 90.24% | 59.37% | 86.28% | 90.30% | 55.47% |
| 2 | 2 days | 44.56% | 60.97% | 25.99% | 51.69% | 68.05% | 32.50% | 55.61% | 60.84% | 29.15% |
| | 4 days | 63.84% | 77.21% | 37.47% | 67.41% | 85.41% | 45.81% | 74.99% | 78.22% | 38.03% |
| | 8 days | 84.09% | 90.37% | 53.03% | 84.56% | 95.56% | 54.99% | 85.63% | 88.27% | 52.50% |
| 3 | 2 days | 50.22% | 58.83% | 35.60% | 49.79% | 63.04% | 49.93% | 56.94% | 65.57% | 45.42% |
| | 4 days | 68.16% | 76.86% | 48.47% | 71.26% | 83.03% | 57.92% | 75.20% | 80.51% | 52.89% |
| | 8 days | 83.80% | 87.99% | 55.32% | 86.47% | 92.85% | 62.85% | 86.38% | 90.98% | 55.35% |

**Table 4.** $NO_3^-$-N concentration reduction rates of in wetlands over 8 days.

| Test No. | Time | Wetland 1 | Wetland with EM 1 | EM-Only 1 | Wetland 2 | Wetland with EM 2 | EM-Only 2 | Wetland 3 | Wetland with EM 3 | EM-Only 3 |
|---|---|---|---|---|---|---|---|---|---|---|
| 1 | 2 days | 45.10% | 62.76% | 39.44% | 51.89% | 58.78% | 30.52% | 52.28% | 71.80% | 31.24% |
| | 4 days | 67.02% | 82.63% | 50.06% | 72.00% | 85.84% | 47.34% | 70.26% | 80.35% | 50.29% |
| | 8 days | 80.10% | 90.75% | 60.68% | 81.02% | 90.68% | 56.26% | 83.08% | 86.32% | 62.02% |
| 2 | 2 days | 51.78% | 65.27% | 35.09% | 61.44% | 73.36% | 50.80% | 52.70% | 70.23% | 37.45% |
| | 4 days | 72.55% | 82.05% | 48.98% | 80.09% | 85.93% | 59.90% | 71.62% | 83.80% | 45.80% |
| | 8 days | 83.19% | 89.63% | 56.88% | 86.43% | 92.62% | 64.31% | 80.15% | 89.05% | 52.09% |
| 3 | 2 days | 53.92% | 63.96% | 40.80% | 46.91% | 63.03% | 37.19% | 44.32% | 69.77% | 42.49% |
| | 4 days | 70.04% | 80.92% | 57.53% | 75.06% | 78.49% | 55.97% | 75.62% | 86.30% | 49.31% |
| | 8 days | 80.88% | 88.27% | 66.07% | 82.24% | 82.16% | 66.23% | 80.92% | 91.74% | 57.43% |

**Table 5.** TP concentration reduction rates in wetlands over 8 days.

| Test No. | Time | Time | Wetland 1 | Wetland with EM 1 | EM-Only 1 | Wetland 2 | Wetland with EM 2 | EM-Only 2 | Wetland 3 | Wetland with EM 3 |
|---|---|---|---|---|---|---|---|---|---|---|
| 1 | 2 days | 14.43% | 30.00% | 3.87% | 15.03% | 20.37% | 14.32% | 5.18% | 18.20% | 13.24% |
| | 4 days | 28.11% | 39.53% | 8.25% | 21.72% | 33.99% | 22.94% | 14.49% | 39.23% | 25.14% |
| | 8 days | 35.92% | 48.91% | 25.23% | 37.62% | 46.87% | 36.59% | 27.17% | 53.16% | 30.35% |
| 2 | 2 days | 8.56% | 27.58% | 9.29% | 14.84% | 21.66% | 18.05% | 15.92% | 18.07% | 16.50% |
| | 4 days | 17.61% | 37.24% | 22.15% | 19.66% | 38.60% | 30.05% | 24.91% | 36.57% | 26.30% |
| | 8 days | 25.12% | 48.35% | 30.75% | 29.83% | 45.45% | 32.36% | 36.43% | 46.00% | 35.45% |
| 3 | 2 days | 21.46% | 31.03% | 21.97% | 14.98% | 26.71% | 18.32% | 24.46% | 30.28% | 21.88% |
| | 4 days | 32.41% | 44.14% | 30.84% | 25.86% | 37.47% | 36.87% | 38.08% | 43.23% | 25.70% |
| | 8 days | 36.80% | 54.15% | 36.62% | 36.26% | 51.56% | 39.52% | 40.22% | 58.12% | 30.96% |

The mean reduction rates are calculated from the reduction rates under three different experimental conditions at three separate test times (i.e., nine data points). After 8 days of treatment, for the EM wetlands, the mean reduction rates of TN, $NH_4^+$-N, $NO_3^-$-N, and TP were 10%, 11%, 11%, and 20% higher than those of non-EM wetlands, respectively, and 42%, 43%, 37%, and 25% higher than those of EM boxes, respectively. After the first 4 days of treatment, for the EM wetlands, the mean reduction rates of TN, $NH_4^+$-N, $NO_3^-$-N, and TP were 14%, 10%, 10%, and 17% higher than those of non-EM wetlands, respectively, and 30%, 32%, 27%, and 17% higher than those of EM boxes, respectively. The difference between the mean reduction rates was statistically analyzed to be significant ($p < 0.05$) by the *t*-test. These show that the effectiveness of treatment by EM wetlands is better than that of non-EM wetlands and EM boxes. Besides being more effective, EM also accelerates the treatment process and thus improves treatment capacity of wetlands rapidly. In the first 4 days during treatment, the concentration of nitrogen and phosphorus decreased sharply, with the concentrations of nitrogen and phosphorus decreasing at the reduction rates of 73–83% and 33–44%, respectively. The concentrations decreased even faster in the first 2 days, with the concentrations of nitrogen and phosphorus decreasing at 50–60% and 20–31%, respectively. Thus, if the rainfall was frequent, EM wetlands could treat drainage water effectively within a short time, and then the receiving water would receive drainage water of good quality.

*3.2. Nitrogen and Phosphorus Loads and Mass Removal Efficiencies in Wetlands*

Nitrogen and phosphorus loads and the mass content of nitrogen and phosphorus in wetlands were calculated from the concentrations multiplied by the volume of water. The water volume was calculated from the water surface area multiplied by the water depth. The water depths were measured at the beginning and the end of each 8-day experiment. The water depth in between (e.g., at the fourth day) was not measured, and it was assumed to drop linearly from the water levels at the beginning and the end of the 8-day experimental period. During each of the three 8-day field experimental periods, the meteorological conditions changed little and there was no rainfall; thus, the linear drop assumption should be acceptable.

The removal efficiency, the decrease in rate of load, was calculated from the difference between the final load and the initial load divided by the initial load. Table 6 shows the removal efficiencies of nitrogen and phosphorus in wetlands. The mean removal efficiencies were calculated from the removal efficiencies under three different experimental conditions at three separate test times (i.e., nine data points). Non-EM wetlands had average removal efficiency of TN and TP of 91% and 73%, respectively. EM boxes had average removal efficiencies of TN and TP of 62% and 53%, respectively. For EM wetlands, TN and TP removal efficiencies of EM wetlands were 95% and 81%, respectively.

EM wetlands had the best treatment effect. The mean TN removal efficiencies were 4% and 33% higher than those of non-EM wetlands and EM boxes, respectively. The mean TP removal efficiencies were 8% and 28% higher than those of non-EM wetlands and EM boxes, respectively. The difference between the mean removal efficiencies was statistically analyzed to be significant ($p < 0.05$) by the *t*-test. Thus, the EM can be considered as an effective enhancement measure for wetlands to reduce non-point pollution from paddy fields.

**Table 6.** Mass removal efficiencies of nitrogen and phosphorus under various experimental conditions.

| Condition | Test No. 1 | | | | | | Test No. 2 | | | | | | Test No. 3 | | | | | |
|---|---|---|---|---|---|---|---|---|---|---|---|---|---|---|---|---|---|---|
| | TN | | | TP | | | TN | | | TP | | | TN | | | TP | | |
| | 0 days (kg ha$^{-1}$) | 8 days (kg ha$^{-1}$) | Removal Efficiency (%) | 0 days (kg ha$^{-1}$) | 8 days (kg ha$^{-1}$) | Removal Efficiency (%) | 0 days (kg ha$^{-1}$) | 8 days (kg ha$^{-1}$) | Removal Efficiency (%) | 0 days (kg ha$^{-1}$) | 8 days (kg ha$^{-1}$) | Removal Efficiency (%) | 0 days (kg ha$^{-1}$) | 8 days (kg ha$^{-1}$) | Removal Efficiency (%) | 0 days (kg ha$^{-1}$) | 8 days (kg ha$^{-1}$) | Removal Efficiency (%) |
| W1 | 26.32 | 2.19 | 91.67 | 3.92 | 0.85 | 78.43 | 16.44 | 1.94 | 88.19 | 4.18 | 1.37 | 67.26 | 27.12 | 2.69 | 90.23 | 3.62 | 0.77 | 78.61 |
| W2 | 23.26 | 2.40 | 89.68 | 3.80 | 1.07 | 71.90 | 15.89 | 1.23 | 92.27 | 4.21 | 1.22 | 71.08 | 25.06 | 1.87 | 92.52 | 3.16 | 0.88 | 72.35 |
| W3 | 28.55 | 2.22 | 92.23 | 4.02 | 1.29 | 68.00 | 15.20 | 1.03 | 93.23 | 4.09 | 0.89 | 78.22 | 25.08 | 2.25 | 91.01 | 3.67 | 0.93 | 74.68 |
| EM1 | 27.39 | 11.68 | 57.36 | 3.66 | 2.05 | 43.92 | 19.10 | 6.07 | 68.21 | 4.46 | 1.96 | 56.05 | 26.14 | 8.51 | 67.44 | 3.94 | 1.64 | 58.51 |
| EM2 | 26.33 | 10.59 | 59.77 | 3.98 | 1.91 | 52.09 | 16.61 | 6.58 | 60.42 | 4.44 | 2.01 | 54.61 | 26.62 | 9.31 | 65.01 | 4.19 | 1.62 | 61.27 |
| EM3 | 27.54 | 11.07 | 59.82 | 4.17 | 2.24 | 46.42 | 17.34 | 7.16 | 58.70 | 4.16 | 2.03 | 51.15 | 25.92 | 9.12 | 64.83 | 3.77 | 1.64 | 56.63 |
| EMW1 | 31.95 | 1.33 | 95.83 | 3.80 | 0.70 | 81.49 | 17.76 | 1.48 | 91.67 | 4.43 | 0.91 | 79.58 | 29.12 | 1.69 | 94.21 | 4.11 | 0.78 | 81.03 |
| EMW2 | 26.75 | 1.63 | 93.92 | 4.15 | 0.91 | 78.09 | 18.06 | 0.76 | 95.77 | 4.44 | 1.11 | 75.01 | 25.92 | 0.86 | 96.68 | 4.25 | 0.75 | 82.28 |
| EMW3 | 29.13 | 1.06 | 96.37 | 4.37 | 0.76 | 82.61 | 18.52 | 0.53 | 97.16 | 3.95 | 0.70 | 82.36 | 26.05 | 1.06 | 95.94 | 3.75 | 0.59 | 84.18 |

The exact mechanisms that EM use to enhance the pollutant removal are not known. There are several stipulations [26,27]. EM could induce the soil to become disease-suppressive in nature by promoting the growth of fungi. These microbial processes could be accelerated, such as through nitrification and denitrification, as well as physicochemical processes such as the fixation of phosphate by iron and aluminum in the soil filter. The rate of decomposition of nitrogen and phosphorus by microorganisms could also be increased. In a better water–soil environment for microorganisms, microorganisms can produce more nutrients by microbial decomposition, which could be taken up and absorbed by plants. The better-grown plants could then intercept more nitrogen and phosphorus compounds. With the possible mechanisms utilized together, the removal rate of nitrogen and phosphorus increase.

Rainfall events of short in-between time gaps occur frequently in Southern China, which may cause frequent drainage/outflow of paddy fields, leading to serious non-point source pollution by the discharged paddy field water, which is rich in nitrogen and phosphorus, into the river. EM wetland technology, a relative rapid biological water treatment method, is recommended for treating paddy field drainage that has a short time detention between storm events.

### 3.3. TN and TP Uptake by Aquatic Vegetables

EM can improve soil quality; enhance the growth, yield, and quality of crops; and reduce the inputs of chemical fertilizers and pesticides in agriculture [28,29].

Table 7 shows yields of aquatic vegetables in wetlands, and Table 8 shows nitrogen and phosphorus uptake of aquatic vegetables.

**Table 7.** Yield of aquatic vegetables in wetlands (kg ha$^{-1}$).

| Vegetable | Wetland 1 | Wetland with EM 1 | Wetland 2 | Wetland with EM 2 | Wetland 3 | Wetland with EM 3 |
|---|---|---|---|---|---|---|
| Water Bamboo | 573.33 | 803.33 | 536.67 | 672.33 | 513.33 | 593.33 |
| Water spinach | - | - | 1526.67 | 1896.67 | - | - |
| Cress | - | - | - | - | 773.33 | 1016.67 |
| Arrowhead | 336.67 | 450.00 | - | - | - | - |

**Table 8.** Content of nitrogen and phosphorus in vegetables (g kg$^{-1}$).

| Vegetable | Wetland 1 | | Wetland with EM 1 | | Wetland 2 | | Wetland with EM 2 | | Wetland 3 | | Wetland with EM 3 | |
|---|---|---|---|---|---|---|---|---|---|---|---|---|
| | TN | TP | TN | TP | TN | TP | TN | TP | TN | TP | TN | TP |
| Water Bamboo | 24.26 | 2.55 | 33.99 | 3.58 | 22.71 | 2.93 | 28.45 | 3.67 | 21.72 | 2.29 | 25.11 | 2.64 |
| Water Spinach | - | - | - | - | 31.78 | 1.57 | 39.48 | 1.95 | - | - | - | - |
| Cress | - | - | - | - | - | - | - | - | 42.34 | 15.66 | 55.66 | 20.59 |
| Arrowhead | 48.00 | 6.93 | 64.15 | 9.27 | - | - | - | - | - | - | - | - |

The results showed EM could improve yields of aquatic vegetables and promote vegetables from taking up additional nitrogen and phosphorus. The average yield of EM wetlands was 21% higher than that of non-EM wetlands, and average TN and TP uptakes by EM wetlands were significantly higher than that of Non-EM wetlands at both 22%, respectively.

## 4. Conclusions

Applying EM into the constructed wetland can accelerate and enhance the concentration reduction of nitrogen and phosphorus in wetlands and improve mass removal efficiencies of nitrogen and phosphorus. After 8 days treatment of drainage water by EM wetlands, the average concentration reduction rates of TN and $NO_3^-$-N in wetlands could reach over 88%, while the reduction rates of $NH_4^+$-N could reach more than 91% and the concentration reduction rate of TP could reach more than 50%. In the first 4 days of treating drainage water by EM wetland technology, the concentration of nitrogen and phosphorus decreased, with a rate higher than that of later days. The concentration reduction rate of TN was 78% and TP was 40% during the first 4 days, which was higher than those

without using EM at 50% and 20%, respectively. Application of EM to wetlands led to additional TN and TP concentration reduction rates of 28% and 20%, respectively.

The mass removal efficiencies of TN and TP in EM-applied wetlands increased by around 4% and around 7%, respectively, when comparing with those of non-EM wetlands, while the mass removal efficiencies of TN and TP in EM-applied wetlands of the first 4 days increased by 10% and around 6%, respectively, when compared with those of non-EM wetlands. Application of EM to wetlands led to TN and TP mass removal efficiencies of 95% and 81%, respectively.

EM can also improve the yield of vegetables by 21%.

**Author Contributions:** Data curation, Q.W.; Formal analysis, X.L.; Funding acquisition, Y.W. and J.X.; Methodology, J.X.; Resources, L.C.; Supervision, L.L.; Writing—original draft, X.L.; Writing—review & editing, Q.G. All authors have read and agreed to the published version of the manuscript.

**Funding:** This research was funded by the National Key Research and Development Program of China (2017YFC1502705), Jiangsu Province Water Conservancy Science and Technology Project (no. 2018065), and Fundamental Research Funds for the Central Universities (no. 2019B17914).

**Acknowledgments:** The authors gratefully acknowledge financial support from China Scholarship Council.

**Conflicts of Interest:** The authors declare no conflict of interest.

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
