# Peer review of "Enhancing Nitrogen and Phosphorus Removal by Applying Effective Microorganisms to Constructed Wetlands"

_water, doi:10.3390/w12092443_

Round 1

Reviewer 1 Report

The study is designed well and conducted. Several aspects of the manuscript are confusing and needs to be clearly discussed.

Please clarify and rewrite the following:

  • What is the control for the study?
  • Non-EM wetlands performance is mentioned several times throughout the results but was not clearly stated what non-EM is.
  • The display and discussion of results are very confusing for example: page 4 lines 133-146, lot of data is shown in one stretch this is just confusing.
  • The performance of non-EM seems to be higher (lines 164-165) that EM wetlands (lines 165-166). In conclusions it is stated EM-wetlands had better performance.
  • The mechanisms of how and why EM can increase the removal of N and P needs be discussed further.
  • Provide evidence of previous laboratory studies or conduct laboratory study.

The graphs and tables represent same data. Move the tables to supporting information and improve the graphs.

Author Response

Point 1: What is the control for the study?

Response 1: The EM boxes and the wetlands (without EM) are the two controls for our study of effectiveness of the wetlands with EM. It would be even better to additionally use the water only as the third control with neither EM or wetland plants.

Point 2: Non-EM wetlands performance is mentioned several times throughout the results but was not clearly stated what non-EM is.

Response 2: Non-EM means the wetlands (with vegetables planted) without EM been applied. We have added an explanation to the article, thank you (line 97).

Point 3: The display and discussion of results are very confusing for example: page 4 lines 133-146, lot of data is shown in one stretch this is just confusing.

Response 3: Thank you for your suggestion, we have removed the long and confusing list of concentration data and only kept the calculated reduction rates in the revised version, and have moved the summary of calculated reduction rates to the next paragraph. (line 226-253)

Point 4: The performance of non-EM seems to be higher (lines 164-165) than EM wetlands (lines 165-166). In conclusions it is stated EM-wetlands had better performance.

Response 4: Concentration reduction rates of the EM wetlands for NH4+-N, NO3--N and TP are 91%, 89% and 50%, respectively. Those of the non-EM wetlands are 80%, 80% and 30%, respectively. Thus, the performance of EM wetlands is better.(line 237-242)

Point 5: The mechanisms of how and why EM can increase the removal of N and P needs be discussed further.

Response 5: Thank you for your advice, the possible mechanisms of EM increase the removal of N and P are added (line 318-327).

Point 6: Provide evidence of previous laboratory studies or conduct laboratory study.

Response 6: Thank you for your suggestion, and the evidence of previous laboratory studies is added. (line 62-75).

Point 7: The graphs and tables represent same data. Move the tables to supporting information and improve the graphs.

Response 7: Thank you for your advice. The graphs and graphs do provide related information but are different. The graphs show the data of concentrations, but the tables show the reduction rates calculated from the concentrations. We provided both the tables and the graphs aiming to provide the original concentration data for the readers if they want to use them later. And we also like the readers see the concentration change rates more clearly. We have changed locations of the tables and graphs to where the supporting information is, making the connections better for the readers.

Reviewer 2 Report

This manuscript mainly discusses the addition of effective microorganisms to improve the removal efficiencies of nitrogen and phosphorous contained pollutant in constructed wetlands. According to the conclusions, effective microorganisms can effectively enhance the removal of total nitrogen, ammonium nitrogen, nitrate-nitrogen, and total phosphorus. However, these conclusions do not examine by the statistical process. In addition, this manuscript is not well prepared for the publication. More detail of the experimental layout shall be described. All the tables and figures shall be rearranged and replotted. All the experimental data does not fully discuss and more discussion about the enhancement of nutrient removal shall be conducted. There are still some minor revisions needed to be finished before considering the possibility of publication.

Minor comments:

P.2, Line 71-73: The description of Figure 1 is quite ambiguous and it is suggested that the authors shall define the type of constructed wetlands, water depth, flow direction, and porosity. Figure 1 shall replot for a better description of the experimental layout. Since the aquatic plants are the same for the experimental and control systems, the duplicate description is no need for both systems.

P.3, Line 95-6: “And” shall be deleted. There is always a blank between the digital numbers and unit. The authors shall explain what is three “time” periods.

P.3 Line 96: There misses a dot between 16th and The.

P.3 Line 105: It is suggested that no aquatic plants shall remark to EM 1-3.

P.4, Line 122-123: “-” and “+” are the superscripts of “NH” and “NO” instead of “4” and “3”.

P.4, Line 128: The way calculating TN and TP uptake by the aquatic plant is missed.

P.4, Line 132: The definition of reduction rate is missed.

P.4, Line 134-136: The authors shall define the “EM-boxed”. The experimental data of TN and TP described in this part cannot be found in Table 2.

P.4, Line 136: “-1” shall be printed in the superscript and similar mistakes are also found in the rest of this manuscript.

P.4, Line 147: The experimental results shall be statistically examined the difference between the experimental system and the control system.

P.4, Line 153: There missed a dot at the end. It is suggested that the authors shall add more theoretic discussion about the mechanism enhancing the removal of TN by the addition of EM.

P.4, Line 154: The explanatory headings of the first two columns are missed. The authors shall redesign all the tables for a better reading of the readers.

P.5, Line 155: The symbols in this figure are not clear enough for the readers. It is suggested to replot. The “/” following TN shall be deleted and the format of the unit, “mg/L” is not the same as they are in the rest of the manuscript.

P.5, Line 160: The sentence is unclear and needs to be rewritten.

P.5, Line 164: The authors shall define the “non-EM”.

P.5, Line 170: The authors shall well explain the meaning of “regulate the nitric and phosphoric nutriment level” and the connection to the removal efficiencies of Nh4+-N, NO3—N, and TP.

P.7, Line 187: The “%” between 3 and – shall be deleted and similar revisions are also needed to be conducted in the rest.

P.8, Line 194: The definition of mass removal efficiency is missed. The authors shall well explain the parameters, “Inlet”, “Removal”, and “Rate”. The data in Table 6 shall be further discussed.

P.8, Line 209: There misses a dot.

P.9, Line 210: The reaction time shall be removed to the remark of Table 6, and the unit, “kg ha-1”, shall move to the explanatory headings. The format of data in Table 6 shall be also defined in the remark.

P.10: The line number is missed after section 3.3. Table 7 and Table 8 shall be rearranged because of forming an obstacle for the reader to understand the meaning of these tables.

P.11: The last paragraph in the conclusion does not belong to the important finding of this manuscript.

  1. 11: Some of the reference formats does not follow the requirement of the Journal.

Author Response

Point 1: P.2, Line 71-73: The description of Figure 1 is quite ambiguous and it is suggested that the authors shall define the type of constructed wetlands, water depth, flow direction, and porosity. Figure 1 shall replot for a better description of the experimental layout. Since the aquatic plants are the same for the experimental and control systems, the duplicate description is no need for both systems.

Response 1: Thank you for your advice. Additional description of the experimental layout is added to make Figure 1 easier to understand (line 89-95, line118-124). And the repetitive description of the aquatic plants is removed (line 97-102).

Point 2: P.3, Line 95-6: “And” shall be deleted. There is always a blank between the digital numbers and unit. The authors shall explain what is three “time” periods.

Response 2: Thank you for your advice. The word ‘And’ is deleted and an explanation of the three ‘time’ periods is added. (line 130-136)

Point 3: P.3 Line 96: There misses a dot between 16th and The.

Response 3: A dot is added, thank you for your careful review. (line 135)

Point 4: P.3 Line 105: It is suggested that no aquatic plants shall remark to EM 1-3.

Response 4: The words “no aquatic plants” were added to EM 1-3, thank you. (Table 1)

Point 5: P.4, Line 122-123: “-” and “+” are the superscripts of “NH” and “NO” instead of “4” and “3”.

Response 5: Thank you for pointing out the typo errors. All the typos of ‘NH4+-N’ and ‘NO3--N’ have been corrected.

Point 6: P.4, Line 128: The way calculating TN and TP uptake by the aquatic plant is missed.

Response 6: Sorry for missing the introduction of the calculation method for TN and TP uptake by aquatic plant, it is now added. (line 170-172)

Point 7: P.4, Line 132: The definition of reduction rate is missed.

Response 7: The definition of reduction rate is now added. (226-228)

Point 8: P.4, Line 134-136: The authors shall define the “EM-boxed”. The experimental data of TN and TP described in this part cannot be found in Table 2.

Response 8: The definition of EM-boxes is added (line 116-117). The original experimental data for concentration are shown in figures. The calculated reduction rates of concentration are shown in tables.

Point 9: P.4, Line 136: “-1” shall be printed in the superscript and similar mistakes are also found in the rest of this manuscript.

Response 9: All the ‘-1’ mistakes in the manuscript are now fixed, thank you for your careful review.

Point 10: P.4, Line 147: The experimental results shall be statistically examined the difference between the experimental system and the control system.

Response 10: Thank you for your great suggestion, differences between the mean concentrations of TN, TP , NH4+-N and NO3--N under different experimental conditions are statistically analyzed by the T-test, and all the T-test results show p<0.05. (line 252-253)

Point 11: P.4, Line 153: There missed a dot at the end. It is suggested that the authors shall add more theoretic discussion about the mechanism enhancing the removal of TN by the addition of EM.

Response 11: Thank you for your advice, the dot is now added, and the possible mechanisms of EM’s increase in the removal of N and P are also now added. (line 318-)

Point 12: P.4, Line 154: The explanatory headings of the first two columns are missed. The authors shall redesign all the tables for a better reading of the readers.

Response 12: The explanatory headings of the first two columns of each table are now added. (Table 2,3,4,5). Symbols in the headings have been replaced by words to make them easier to read. Thank you for your good advice.

Point 13: P.5, Line 155: The symbols in this figure are not clear enough for the readers. It is suggested to replot. The “/” following TN shall be deleted and the format of the unit, “mg/L” is not the same as they are in the rest of the manuscript.

Response 13: Thank you for your advice. All the figures are now replotted by changing the color of lines to make them easier to distinguish. All the ‘/’ symbols are now deleted,and ‘mg/L’ are now changed into mg l-1

Point 14: P.5, Line 160: The sentence is unclear and needs to be rewritten.

Response 14: The sentence is unnecessary, so it is deleted. (line 212-214)

Point 15: P.5, Line 164: The authors shall define the “non-EM”.

Response 15: The definition of ‘non-EM’ is now added and the descriptive words added. (Line 93)

Point 16: P.5, Line 170: The authors shall well explain the meaning of “regulate the nitric and phosphoric nutriment level” and the connection to the removal efficiencies of Nh4+-N, NO3—N, and TP.

Response 16: We are sorry for using the wrong words. Explanation of the meaning is now added in the discussion part. (Line 245-254)

Point 17: P.7, Line 187: The “%” between 3 and – shall be deleted and similar revisions are also needed to be conducted in the rest.

Response 17: ‘%’ between numbers and ‘-’  are now deleted from the entire manuscript.

Point 18: P.8, Line 194: The definition of mass removal efficiency is missed. The authors shall well explain the parameters, “Inlet”, “Removal”, and “Rate”. The data in Table 6 shall be further discussed.

Response 18: Thank you for your suggestion. The definition of removal rate is added (line 293-297). And the ‘Inlet’, ‘Removal’, and ‘Rate’ are now changed to ‘0 day’, ‘8 days’ and removal efficiency to make them easier for the readers to understand. Additional discussion has also been added (line 298-309).

Point 19: P.8, Line 209: There misses a dot.

Response 19: A dot is added.

Point 20: P.9, Line 210: The reaction time shall be removed to the remark of Table 6, and the unit, “kg ha-1”, shall move to the explanatory headings. The format of data in Table 6 shall be also defined in the remark.

Response 20: Thank you for your advices. The reaction time is removed from the table title and the unit ‘kg ha-1’ is moved to the explanatory headings.

Point 21: P.10: The line number is missed after section 3.3. Table 7 and Table 8 shall be rearranged because of forming an obstacle for the reader to understand the meaning of these tables.

Response 21: Sorry for missing the line number in the previously submitted manuscript possibly due to problem with the online conversion from WORD document to PDF. Table 7 and Table 8 are rearranged.

Point 22: The last paragraph in the conclusion does not belong to the important finding of this manuscript.

Response 22: Thank you for your suggestion. This paragraph is moved to discussion section (line 332-337).

Point 23: Some of the reference formats does not follow the requirement of the Journal.

Response 23: Thank you for your careful review, all reference formats are checked and modified.

Round 2

Reviewer 1 Report

Authors addressed previous comments and improved the manuscript

Author Response

Thank you for your all suggestion and patient.

Reviewer 2 Report

The manuscript had been revised by the authors. However, there are still some minor revisions needed to be finished before considering the possibility of publication. The line number indicated in the response seems to be wrong. Some of them can be easily identified by the reviewer, but, I’m not sure about the related content. It is strongly suggested that the authors shall carefully check the manuscript before the submission.

Minor comments:

P.2, Line 62: The abbreviation of nomenclature, “TN” and “TP”, shall be clearly defined when it appears the first time.

P.3, Line 116-117: There is always a blank between the digital number and unit. The definition of “EM-boxed” is not found at Line 116-117.

P.4, Line 144: Duplicate definitions of  “NH4 +-N)” and “NO3 --N” are found at Line 67.

P.4, Line 170-172: The way calculating TN and TP uptake by the aquatic plant is not found at Line 170-172.

P.4, Line 226-228: The definition of reduction rate is not found at Line 226-228.

P.4, Line 252-253: The results of the statistical examination are not found at Line 252-253. Is the description at Line 232-243 the related results?  The “p” shall be printed in italic and the “T-test” shall be printed as “t-test”.

P.4, Line 318: The theoretic discussion about the mechanism enhancing the removal of TN by the addition of EM is not found at line 318.

Author Response

Thank you for your great advice, and your patient. And sorry for marking wrong line numbers in the former revision.

Point 1: P.2, Line 62: The abbreviation of nomenclature, “TN” and “TP”, shall be clearly defined when it appears the first time.

Response 1: Thank you for your advice, the definition of ‘TN’ and ‘TP’ are added. (Line 62)

Point 2: P.3, Line 116-117: There is always a blank between the digital number and unit. The definition of “EM-boxed” is not found at Line 116-117.

Response 2: A blank between ‘50’ and ‘mm’ is added, thank you for your suggestion. And the definition of ‘EM-box’ is at Line 107-111.

Point 3: P.4, Line 144: Duplicate definitions of “NH4 +-N” and “NO3 --N” are found at Line 67.

Response 3: Thank you for your careful review, duplicate definitions are deleted. (Line 143)

Point 4: P.4, Line 170-172: The way calculating TN and TP uptake by the aquatic plant is not found at Line 170-172.

Response 4: The way calculating TN and TP uptake by the aquatic plant is at Line 148-150.

Point 5: P.4, Line 226-228: The definition of reduction rate is not found at Line 226-228.

Response 5: The definition of reduction rate is at Line 170-172.

Point 6: P.4, Line 252-253: The results of the statistical examination are not found at Line 252-253. Is the description at Line 232-243 the related results?  The “p” shall be printed in italic and the “T-test” shall be printed as “t-test”.

Response 6: Sorry for writing wrong line numbers. The results of the statistical examination are at Line 230-232. And ‘p’ is printed in italic, ‘T-test’ is printed as ‘t-test’ also.

Point 7: P.4, Line 318: The theoretic discussion about the mechanism enhancing the removal of TN by the addition of EM is not found at line 318.

Response 7: The possible mechanisms of EM increase the removal of N are at Line 234-243.